# Oxcarbazepine and Hyponatremia

**DOI:** 10.3390/medicina58050559

**Published:** 2022-04-19

**Authors:** Julija Čiauškaitė, Giedrė Gelžinienė, Giedrė Jurkevičienė

**Affiliations:** Department of Neurology, Medical Academy, Lithuanian University of Health Sciences, A. Mickevičiaus Str. 9, LT 44307 Kaunas, Lithuania; giedre.gelziniene@lsmu.lt (G.G.); giedre.jurkeviciene@lsmu.lt (G.J.)

**Keywords:** oxcarbazepine, hyponatremia, epilepsy

## Abstract

*Background and Objectives:* Hyponatremia is one of the most common adverse effects in patients treated with oxcarbazepine (OXC). Different risk factors for OXC-induced hyponatremia have been described as age, female gender, dosage, and combination with other drugs During our clinical practice, we noticed that a longer duration of treatment with OXC could be associated with a higher risk of hyponatremia, therefore, in this study, we aimed to evaluate factors that may increase the risk of OXC-induced hyponatremia. *Materials and Methods:* Data were retrospectively collected from our clinical database at the Department of Neurology of the Hospital of Lithuanian University of Health Sciences Kaunas Clinics. The sample was divided into three groups: OXC consumers (*n* = 31), other anti-seizure medications (ASMs) consumers (*n* = 43), and controls absent ASMs (*n* = 31). All groups were matched by age and gender. Hyponatremia was defined as <136 mmol/L. *Results:* The frequency of hyponatremia was significantly higher among OXC patients (61.3%) compared to other ASM patients (5.4%) and controls (3.2%). The mean serum sodium concentration in the OXC group was 133.1 ± 5.1 mmol/L. The frequency of severe hyponatremia among OXC-treated patients was 19.4%; this subgroup was older than patients with moderate hyponatremia and normonatremia and had a longer OXC treatment duration compared to a subgroup of normonatremia. The average duration of OXC therapy was 8.7 ± 5.5 years with a range from 1 to 21 years. Serum sodium concentration and duration of treatment with OXC demonstrated a significant negative correlation (r = −0,427, *p* = 0.017). Each year of therapy with OXC increased the risk of hyponatremia 1.3 times (OR = 1.326, 95% Cl 1.027–1.712, *p* = 0.031). Other factors (gender, age, polypharmacy, OXC dosage, and serum concentration) did not show a significant association with the development of hyponatremia. *Conclusions:* Longer duration of treatment with OXC is an important factor in the development and severity of hyponatremia.

## 1. Introduction

Oxcarbazepine (OXC) is an oral drug for the treatment of focal onset epilepsy in both monotherapy and adjunctive therapy [1]. Together with carbamazepine (CBZ) and eslicarbazepine acetate, OXC is a member of the dibenzazepine carboxamide class of putative voltage-gated sodium channel blocking anti-seizure medications (ASMs) [2]. Hyponatremia may develop with all drugs of this class but is more frequent with OXC [3]. It is often asymptomatic, but it can lead to an increased seizure frequency, respiratory distress, and even coma in severe cases [4]. The prevalence of hyponatremia varies from 17.9% to 73.3% in OXC-treated patients [4,5].

Possible pathophysiology mechanisms of this adverse effect include a direct effect of OXC on the renal collecting tubules or an enhancement of their responsiveness to circulating vasopressin [6].

Vasopressin activates the production of cyclic adenosine monophosphate (cAMP). Through intermediate mechanisms, cAMP activates aquaporin migration to the apical plasma membrane and increases water permeability. Prostaglandin E2 (PGE2) inhibits this process by reducing the production of cAMP and OXC inhibits PGE2 formation. As a result, renal water absorption increases whereas osmolarity and sodium concentration decrease [7].

Different risk factors for OXC-induced hyponatremia have been described. Previous studies demonstrated that old age or OXC in combination with other drugs increased the risk of hyponatremia [4,8]. Lin et al. reported that the use of OXC in patients with epilepsy is associated with a dose-dependent reduction in serum sodium levels [9]. A genetic predictor of OXC-induced hyponatremia has not yet been identified [10]. During our clinical practice, we have noticed that a longer duration of treatment with OXC could be associated with a higher risk of hyponatremia, therefore, in this study, we aim to evaluate the factors that may increase the risk of OXC-induced hyponatremia.

## 2. Materials and Methods

Retrospective data were collected from our clinical database at the Department of Neurology of the Hospital of Lithuanian University of Health Sciences Kaunas Clinics. We assessed cases of patients who had visited from 1 January to 31 December in 2021. The sample was divided into three groups: OXC (mono- or polytherapy) consumers (*n* = 31), other ASMs (carbamazepine, valproic acid, topiramate, lamotrigine, or levetiracetam) consumers (*n* = 43), and controls absent ASM (*n* = 31). Controls had newly diagnosed or suspected epilepsy and underwent routine laboratory tests before ASM prescription. All groups were matched by age and gender.

Patients who met the following criteria were included in this study: adults with a definite or suspected diagnosis of epilepsy and case records containing clinical information: use of ASMs and other medication, dosage, duration of ASM therapy (duration of ASM therapy was determined from the first prescription to the visit when their laboratory tests were performed), serum sodium, creatinine, glycemia, ASM concentrations, and concomitant diseases. Serum levels of ASM in our clinical laboratory are assayed by liquid chromatography, and the reference range of OXC concentration is 10–35 µg/mL.

Individuals who had other conditions that could potentially affect sodium levels (e.g., pregnancy, use of diuretics, adrenal gland insufficiency and hypopituitarism, and kidney failure) were excluded. Hyponatremia was defined as a sodium concentration lower than 136 mmol/L, instances between 128 and 136 mmol were described as moderate hyponatremia, and levels lower than or equal to 128 mmol/L meant severe hyponatremia.

The following data were collected: demographic information (gender, age) and clinical characteristics (ASMs, sodium concentration, duration of ASM therapy).

The study design was approved by the Ethics Committee for Biomedical Research at the Lithuanian University of Health Sciences (No BE-2-107, 23 November 2020), Kaunas, Lithuania.

### Statistical Analysis

Data for continuous variables were tested for normality using the Shapiro–Wilk test. For the comparisons of more than two groups, we carried out a parametric and nonparametric analysis of variance (ANOVA and Kruskale-Wallis tests). Games-Howell post hoc tests were used for multiple comparisons. Data of the two groups were compared using the Mann–Whitney U test. Chi-square tests for categorical variables were used. A binary logistic regression analysis was conducted to estimate risk factors for hyponatremia. Spearman’s rank correlation coefficient was used to measure the strength of the association.

The level of statistical significance for testing statistical hypotheses was 0.05.

The statistical analysis was carried out on SPSS version 23.0 for Windows.

## 3. Results

### 3.1. Clinical and Demographic Characteristics

The clinical and demographic characteristics of patients who used OXC, other ASMs, and control subjects are summarized in Table 1. The age and gender distribution of all three groups did not differ significantly. Glycaemia ranges were from 4.53 to 6.13 mmol/L and creatinine ranges were from 57 to 92 μmol/L. There was no notable difference in the duration of ASM therapy and frequency of polytherapy between OXC and other ASM groups. The average daily dose of OXC was 1014.5 ± 430.1 mg (range from 300 to 1950 mg) and the serum concentration of OXC was 15.4 ± 6.3 µg/mL (range from 3.8 to 28.8). OXC as monotherapy was used by eight (25.8%) participants. The average duration of OXC therapy was 8.7 ± 5.5 years ranging from 1 to 21 years.

The mean sodium concentration in other ASMs and control groups was within the normal range and differed from the OXC group where sodium concentration was abnormally low.

The frequency of hyponatremia was higher among OXC patients (61.3%) compared to other ASMs patients (5.4%) and controls (3.2%), *p* < 0.005.

### 3.2. Hyponatremia and Clinical Factors

The comparative results of the clinical features between OXC-treated patients with and without hyponatremia are listed in Table 2. Duration of OXC therapy was longer among patients with hyponatremia compared to eunatremia (*p* < 0.05), whereas average age, dosage, and OXC concentration in blood serum did not differ. The distribution of gender and poly-mono therapy did not differ either.

Serum sodium concentration and duration of treatment with oxcarbazepine demonstrated a significant negative correlation (r = −0.427, *p* = 0.017). Other investigated factors are reported in Table 2 and Table 3.

The comparative results of the clinical features between OXC-treated patients with severe, moderate hyponatremia and normonatremia are listed in Table 4. Statistical analysis of the three subgroups revealed that the patients with severe hyponatremia were significantly older compared to the other two groups. The duration of treatment was also longer compared to the patients with normonatremia (*p* < 0.05).

We analyzed the data with binary logistic regression analysis to investigate the likely variables that lead to the development of hyponatremia. The analysis demonstrated that a single year of therapy with OXC increased the risk of hyponatremia 1.3 times. Gender, age, dosage, serum concentration of OXC, and polypharmacy were the other investigated risk factors for the development of hyponatremia, these are shown in Table 5. The percentage of correct predictions was 89.5%.

## 4. Discussion

ASM treatment is associated with numerous side effects. One such unfavorable reaction is hyponatremia.

Lower serum sodium concentration may develop with the use of carbamazepine, eslicarbazepine, sodium valproate, lamotrigine, levetiracetam, however, among all ASMs, OXC-induced hyponatremia is the most frequent [11].

In our study more than one-fifth of all participants had abnormally low sodium concentration values. The frequency of hyponatremia was greater in the OXC group compared to the other ASMs or the control group. Lower serum concentration was estimated in more than half (61.3%) of OXC-receiving patients and this incidence is slightly higher than that demonstrated in previous studies (39.4–51.2%) of similar sample sizes [12,13]. Severe hyponatremia was found in 19.4% of the OXC group, for comparison, in other studies it ranged from 11.1% to 22% [4,8]. The average serum sodium levels among other ASMs and control groups were in the normal range (138.7 ± 2.3 and 138.9 ± 1.5, respectively) while in the OXC group, sodium concentration was below the normal range (133.1 ± 5.1). The fact that the average sodium concentration of patients who used OXC as monotherapy or in combination with other ASMs did not differ, but was lower compared to other ASM groups, suggests that hyponatremia was induced by OXC therapy.

Data on risk factors for OXC-induced hyponatremia are limited and results vary. For instance, the results of several previous studies showed that older age is associated with a higher risk of hyponatremia [4,8,14]. Meanwhile, our data do not fully support these findings; patients with hyponatremia were only slightly older than those with normonatremia and the difference was not significant. However, subgroup analysis showed that OXC-treated patients with severe hyponatremia were significantly older compared to patients with normonatremia and moderate hyponatremia (62.3 ± 6.3 vs. 43.4 ± 16.5 vs. 43.8 ± 14.7 years). This suggests that the influence of older age may be more pronounced in the development of severe hyponatremia, although a larger sample size is needed to make a more reliable conclusion.

There are some data suggesting that female patients on OXC were at higher risk for hyponatremia than male patients [8,15]. In our study, distribution of gender did not differ between normonatremia and hyponatremia subgroups.

Nielsen et al. discovered that patients treated with dosages above 30 mg/kg/day had a significantly higher risk of becoming hyponatremic [13]. Lin et al. also found a relationship between the dosage of OXC and the presence of hyponatremia [9]. Although the mean dosage of OXC in the recently mentioned study and ours was similar (1000.7 ± 439.5 vs. 1014.5 ± 430.1 mg), our result did not confirm an association between the OXC dosage and the risk of hyponatremia; the influence of OXC concentration was also not established.

Our data showed that the duration of OXC therapy was significantly longer among patients with hyponatremia compared to those with eunatremia. This difference was even greater after splitting patients into groups with severe and normonatremia. Serum sodium concentration had a significantly negative correlation with the duration of OXC therapy.

In a large cohort study, the influence of OXC treatment duration on symptomatic hyponatremia was not established [4]. It can be noted that the average age of patients with severe hyponatremia in this study was very similar to our findings (62.35 ± 15.5 vs. 62.3 ± 6.3). The mean duration of OXC treatment was 1280 ± 1236 days which would approximately correspond to 3.5 ± 3.3 years, whereas in our study it was markedly higher—8.7 ± 5.5 years. In summary, these results could lead us to assume that despite the large sample in the previous study, the influence of longer OXC therapy for hyponatremia development could not emerge due to a short period of OXC consumption among participants. However, in the Marinez et al. study, 203 patients were observed for 16 weeks after OXC initiation and 15% (*n* = 31) of participants’ low serum sodium values (<135 mmol/L) were established [16]. Their results show that hyponatremia also has an early onset adverse effect. More studies are needed to clarify these observations.

From all investigated factors, only the duration of treatment with OXC significantly increased the risk of developing hyponatremia: one year of therapy with OXC increased the risk of hyponatremia 1.3 times.

In most cases, epilepsy requires long-term treatment, therefore, it is important to be aware of possible adverse effects of ASMs, which may appear over time. The lower sodium concentration is often assumed to be asymptomatic, but it can lead to various symptoms such as headache, irritability, nausea or vomiting, mental slowing, confusion, or disorientation. Significant hyponatremic symptoms include seizure aggravation or respiratory distress [4,17]. Some of the symptoms that are effectively considered as side effects of OXC may actually be caused by hyponatremia. This hypothesis was analyzed in a previous study where 1370 patients treated with CBZ or OXC were identified, 410 of whom had at least one episode of hyponatremia. Authors found that adverse effects (dizziness, tiredness, instability, and diplopia) occurred in 65% of participants with hyponatremia compared to 21% with normal sodium levels (odds ratio 7.5, *p* ≤ 0.001) and in 83% of people with severe hyponatremia compared to 55% in those with mild hyponatremia (*p* ≤ 0.001) [18]. Previous studies also found that even mild chronic hyponatremia can cause neurocognitive deficits and gait disturbance falls, and the treatment of this disturbance of electrolytes improved neurocognitive and neuromuscular function [19,20]. Therefore, clinicians should provide regular monitoring of serum sodium levels for patients who use OXC and be aware that the risk of hyponatremia may increase over time. In the present study, we did not collect clinical data about possible symptoms of hyponatremia, while the retrospective design may cause a lack of accuracy. Another limitation is the small sample size which reduces the power of the study and increases the margin of error.

## 5. Conclusions

Hyponatremia is a common problem in patients taking OXC, therefore regular ascertainment of sodium levels is recommended. Based on the data from our study, a longer duration of treatment with OXC is an important factor in the development and severity of hyponatremia. OXC-treated patients with severe hyponatremia were significantly older compared to patients with normonatremia and moderate hyponatremia. Additional prospective studies are necessary to confirm these findings.

## Figures and Tables

**Table 1 medicina-58-00559-t001:** Clinical and demographic characteristics.

Characteristic	Total (*n* = 105)	OXC Group (*n* = 31)	Other ASM Group (*n* = 43)	Control Group (*n* = 31)
Age (y), m ± SD	45.7 ± 13.9	47.26 ± 15.8	42.1 ± 11.4	49.39 ± 14.3
Gender				
Male, *n* (%)	44 (41.9%)	11 (35.5%)	16 (37.2%)	17 (54.8%)
Female, *n* (%)	61 (58.1%)	20 (64.5%)	27 (62.8%)	14 (45.2%)
ASMs therapy (*n* = 74)				
Monotherapy, *n* (%)	25 (33.8%)	8 (25.8%)	17 (39.5%)	-
Polytherapy, *n* (%)	49 (66.2%)	23 (74.2%)	26 (60.5%)	-
Duration of ASM therapy (y), m ± SD	7.7 ± 4.7	8.7 ± 5.5	6.9 ± 3.8	-
Serum sodium concentration mmol/L, mean ± SD	137.1 ± 4.1	133.1 ± 5.1 *	138.7 ± 2.3	138.9 ± 1.5
Monotherapy	136.2 ± 4.4	132.5 ± 6.6 *	138.2 ± 2.3	-
Polytherapy	136.4 ± 4.7	133.3 ± 5.0 *	139.0 ± 2.0	-
Male	137.5 ± 4.0	134.0 ± 3.4 *	139.7 ± 2.1	139.1 ± 1.3
Female	135.7 ± 4.8	132.7 ± 5.8 *	138.0 ± 2.1	138.7 ± 1.7
Hyponatremia, *n* (%)	23 (21.9%)	19 (61.3%) *	3 (7.0%)	1 (3.2%)

* *p* < 0.05, OXC compared to other ASMs and the control group. OXC—Oxcarbazepine; ASMs—anti-seizure medications; SD—standard deviation.

**Table 2 medicina-58-00559-t002:** Comparisons of demographic and clinical features between OXC- treated patients with and without hyponatremia.

	OXC Group (*n* = 31)			
	Total Group (*n* = 31)	Eunatremia (*n* = 12)	Hyponatremia (*n* = 19)	*p* Values
Age (y), m ± SD	47.26 ± 15.8	43.4 ± 16.5	49.7 ± 15.2	0.287
Gender				
Male (N = 11), *n* (%)	11 (35.5%)	4 (36.4%)	7 (63.6%)	0.577
Female (N = 20), *n* (%)	20 (64.5%)	8 (40.0%)	12 (60.0%)	
OXC therapy				
Monotherapy (N = 8), *n*, %	8 (25.8%)	3 (37.5%)	5 (62.5%)	0.638
Polytherapy (N = 23), *n*, %	23 (74.2%)	9 (39.1%)	14 (60.9%)	
Dosage mg/day, m ± SD	1014.5 ± 430.1	962.5 ± 449.8	1047.3 ± 426.3	0.562
Serum concentration µg/mL, m ± SD	15.4 ± 6.3	14.2 ± 5.8	16.1 ± 6.6	0.306
Duration (y), m ± SD	8.7 ± 5.5	5.6 ± 4.8	10.7 ± 5.1	0.018

OXC—Oxcarbazepine; SD—standard deviation.

**Table 3 medicina-58-00559-t003:** Correlations between clinical factors and serum sodium concentration among OXC- treated patients (N = 31).

Factor	r	*p*
Age in years	−0.294	0.109
Dosage of OXC mg/day	0.288	0.116
Serum concentration of OXC µg/mL	−0.300	0.101
Duration of OXC therapy in years	−0.427	0.017

OXC—Oxcarbazepine.

**Table 4 medicina-58-00559-t004:** Comparisons of demographic and clinical data between OXC- treated patients with severe, moderate hyponatremia and normonatremia.

	Na ≤ 128 mmol/L	128 < Na < 136 mmol/L	Na ≥ 136 mmol/L
Overall (N = 31)	6 (19.4%)	13 (41.9%)	12 (38.7%)
OXC therapy			
Monotherapy (N = 8)	2 (25.0%)	3 (37.5%)	3 (37.5%)
Polytherapy (N = 23)	4 (17.4%)	10 (43.5%)	9 (39.1%)
Gender			
Male (N = 11)	0	7 (63.6%)	4 (36.4%)
Female (N = 20)	6 (30%)	6 (30%)	8 (40%)
Age (y)			
m ± SD	62.3 ± 6.3 *	43.8 ± 14.7	43.4 ± 16.5
median (range)	64 (51–68)	50 (20–61)	37 (27–77)
Dosage of OXC mg/day			
m ± SD	1150 ± 122.4	1000 ± 508.6	962.5 ± 449.8
median (range)	1200 (900–1200)	900 (300–1800)	750 (450–1950)
Serum concentration of OXC			
µg/mL, m ± SD	19.2 ± 5.3	14.6 ± 6.8	14.2 ± 5.8
median (range)	19.1 (12.5–27.6)	14.59 (3.8–26.8)	12.98 (8.2–28.8)
Duration of OXC therapy (y)			
m ± SD	13.2 ± 5.4 #11 {8–20}	9.5 ± 4.7	5.6 ± 4.9
median (range)		8 (2–21)	2.5 (1–15)

* *p* < 0.05, comparison between OXC-treated patients with severe hyponatremia compared to subgroups of moderate hyponatremia and normonatremia. # *p* < 0.05, comparison between OXC-treated patients with severe hyponatremia compared to a subgroup of normonatremia. OXC—Oxcarbazepine; SD—standard deviation.

**Table 5 medicina-58-00559-t005:** Analysis of effective variables that lead to the development of hyponatremia among patients who take OXC.

Factor	Exp (B)	95% Cl	*p*
Age (each additional year)	0.988	0.922–1.059	0.737
Gender (man)	1.723	0.191–15.507	0.628
Duration of therapy (each additional year)	1.326	1.027–1.712	0.031
Dosage of OXC mg/day	0.998	0.993–1.004	0.572
Serum concentration of OXC µg/mL	1.200	0.784–1.837	0.401
Polytherapy	1.617	0.169–15.463	0.278

## Data Availability

The data that support the findings of this study are available from the corresponding author upon reasonable request.

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
