# Peer review of "Oxcarbazepine and Hyponatremia"

_medicina, 2022, doi:10.3390/medicina58050559_

Round 1
Reviewer 1 Report
Thank you for your good manuscript. Please check the attached file

Author Response
- We added the full terms in abstract.
- We corrected numbers in page 4 line 140 (commas were changed to periods).
- In table 3 word potassium was changed to sodium.
- Indeed, line 162 on page 5 had the wrong table number provided, we corrected number 3 to 4.
- We agree that the clinical impact is very important aspect. However we did not collect additional clinical data on hyponatremic patients. Due to the retrospective design of the study we could not collect data on the possible symptoms of hyponatremia, as we could not confirm if the information in medical records would be detailed enough and making additions was not an option. In medical records no other complaints were reported except related to frequency of seizures. That would suggest that most of the hyponatremia cases were chronic and do not cause severe symptoms. Additional prospective studies are necessary to make reliable conclusions. In relation to additional therapy, it was recommended for most of patients with mild hyponatremia to increase dietary salt intake and and to repeat sodium blood test in a few weeks. In cases where hyponatremia was severe, OXC dosage was decreased, and other anti seizure medication was started with the aim to replace OXC.
- We collected medication history data from medical records. Overall the mean age of our partisipants was 45.7±13.9 years, with the elderly consisting a minor part of the sample and expressed polytherapy was uncommon. Consumption of diuretics was one of the exclusion criteria. Other used medications, as far as we know, did not increase the risk of hyponatremia i.e.- metoprolol or alprazolam. In other ASMs group there were a few patient who used antidepresants wich in rare cases could cause lower sodium concentration, but in our study sodium levels of these patients were not below the normal range. That is why we think that this factor did not influence our final results.

Reviewer 2 Report
Dear authors.
This is an interesting case-control study which evaluates the impact of oxcarbazepine in hyponatremia. I found some issues that you should improve.
Methodology
Authors do not describe the time of follow up of their patients nor the total o cases assessed during this time
The n is small. However, depending on the time of follow up this could be acceptable.
Hyponatremia was not corrected with glycemia. Since diabetes was not a exclusion criterion and that mean sodium level in the case-group was mild, it is important to avoid a bias of selection/definition regarding the hyponatremia.
Results
I think it is relevant that author add p value in tables when comparing groups.
Table 3 reports correlation between variables with Serum POTASSIUM or SODIUM? The title refers to POTASSIUM. Correct it.
Table 3 is wrongly referenced in the paragraph below Table 3 in results. I think authors refers to Table 4. Correct it
I suggest performing a multivariable analysis in order to obtain more potent results.
Discussion.
Authors defends their results as compared to previous larger study (refence 1). In order to support their hypothesis, I suggest performing a comparative between those with a treatment below to 3.5 years vs those with a longer time of treatment
Authors conclude that “A longer duration of OXC- therapy is as-sociated with a higher risk and severity of hyponatremia”. I think it is prudent they not be so categorical since a multivariable analysis was not done.
Author Response
Response to Reviewer 2
Methodology
- We assessed cases of patients who had visits in 2021 from January 1st to December 31st. Duration of ASMs therapy was determined from first prescription to the visit when their laboratory tests were performed. Data was based on available medical records.
- We noticed that OXC was prescribed less freequently when compared to other ASMs. Other reasons for the small sample size is due to the lack of information in medical records (ASMs starting data absent, laboratory test not full, ASMs concentration absent), part of the OXC-treated patients were excluded.
- The note for diabetes mellitus was very accurate. We have rewieved our data again and found that one patient in other ASMs group had diabetes. Sodium levels of this patient were 138 mmol/l and glycaemia was 6.13 mmol/l. This might suggest that failure to exclude patients with diabetes did not changed our results significantly. Overall glucose levels were measured for all patients at the same tyme as sodium levels. Glycaemia ranges were from 4.53 to 6.13 mmol/l (we added this information in line 107 at page 3).
Results
- We added p values to Table 2. We added notes in other tables (1 and 4) due to lack of space.
- In table 3 the word potassium was changed to sodium (page 5).
- Reference to Table 3 was corrected to Table 4 (162 line at page 5).
- To investigate the likely variables that lead to the development of hyponatremia, we analyzed data with binary logistic regression analysis (180-181 lines in page 6). As far as we know, logistic regression is one of the techniques of multivariable analysis. Could you please specify which statistical method you had in mind?
Discussion.
- As you suggested, we split OXC group in to those who used OXC for less than 3.5 years (n=8) and those with a longer time of treament (n=23). Patients with longer duration of OXC treatment had lower sodium concentrations (mean±SD respectively 131.78±4.9 and 137.13±3.5, p=0.007).
- Sentence in page 8 lines 261-262 was changed to: ,,Based on the data from our study, a longer duration of treatment with OXC is an important factor in the development and severity of hyponatremia. Additional prospective studies are necessary to confirm these findings.”
